# The Regulation of Collagen Processing by miRNAs in Disease and Possible Implications for Bone Turnover

**DOI:** 10.3390/ijms23010091

**Published:** 2021-12-22

**Authors:** Tomasz P. Lehmann, Urszula Guderska, Klaudia Kałek, Maria Marzec, Agnieszka Urbanek, Alicja Czernikiewicz, Maria Sąsiadek, Paweł Karpiński, Andrzej Pławski, Maciej Głowacki, Paweł P. Jagodziński

**Affiliations:** 1Department of Biochemistry and Molecular Biology, Poznan University of Medical Sciences, 60-781 Poznan, Poland; uguderska@gmail.com (U.G.); klaudiakalek98@gmail.com (K.K.); mariamarzec80@wp.pl (M.M.); aga43221@gmail.com (A.U.); alicja.wojtasiak96@gmail.com (A.C.); pjagodzi@ump.edu.pl (P.P.J.); 2Department of Genetics, Wroclaw Medical University, 50-368 Wroclaw, Poland; maria.sasiadek@umed.wroc.pl (M.S.); polemiraza@poczta.fm (P.K.); 3Institute of Human Genetics, Polish Academy of Sciences, 60-479 Poznan, Poland; andp@man.poznan.pl; 4Department of Paediatric Orthopaedics and Traumatology, Poznan University of Medical Sciences, 61-545 Poznan, Poland; glowackimaciej@o2.pl

**Keywords:** miRNA, extracellular matrix, collagen type I, bone turnover, non-collagenous proteins

## Abstract

This article describes several recent examples of miRNA governing the regulation of the gene expression involved in bone matrix construction. We present the impact of miRNA on the subsequent steps in the formation of collagen type I. Collagen type I is a main factor of mechanical bone stiffness because it constitutes 90–95% of the organic components of the bone. Therefore, the precise epigenetic regulation of collagen formation may have a significant influence on bone structure. We also describe miRNA involvement in the expression of genes, the protein products of which participate in collagen maturation in various tissues and cancer cells. We show how non-collagenous proteins in the extracellular matrix are epigenetically regulated by miRNA in bone and other tissues. We also delineate collagen mineralisation in bones by factors that depend on miRNA molecules. This review reveals the tissue variability of miRNA regulation at different levels of collagen maturation and mineralisation. The functionality of collagen mRNA regulation by miRNA, as proven in other tissues, has not yet been shown in osteoblasts. Several collagen-regulating miRNAs are co-expressed with collagen in bone. We suggest that collagen mRNA regulation by miRNA could also be potentially important in bone metabolism.

## 1. Introduction

Collagen fibres, particularly type I, which is the dominant type of bone collagen, are responsible for the strength and elasticity of bone tissues [1,2]. The process of collagen formation requires intracellular and extracellular and enzymatic and non-enzymatic stages, which have been well described in bone tissue (Table 1). The well-known enzyme-dependent steps of collagen formation are regulated by tissue-specific miRNA, targeting *COL1A1* mRNA (collagen type I gene) and several other mRNA encoding factors relevant to collagen formation. mRNA profiles in tissues are not the main determinants of cell phenotype, since only 19.3% of the total mRNA is tissue specific [3]. Cava et al. found that 39.7% of miRNAs are tissue specific, proving miRNA critical role as regulators of tissue phenotype [3].

MicroRNAs (miRNAs) are small endogenous RNAs with a length of 19–25 nucleotides that regulate gene expression post-transcriptionally, inhibiting their translation or enhancing their degradation. The repression mode that dominates in humans acts without slicing the mRNA and does not require extensive pairing to the miRNA. The deadenylases shorten the poly(A) tail, which in most systems causes mRNA destabilisation through decapping and exonucleolytic decay [4]. miRNAs are involved in essential processes for the development and functioning of the cell, such as cell division, differentiation and programmed death. They regulate the expression of genes at the post-transcriptional level, ultimately affecting the number of individual proteins in the body and ensuring the proper course of processes in the cell. Malfunctions or mutations in the genes encoding miRNAs can lead to serious disorders [4].

Therefore, in this review, we aimed to show how miRNAs regulate collagen and non-collagenous genes in various tissues. Understanding the miRNA-dependent systems that regulate collagen in multiple tissues could be significant in studying similar miRNA–mRNA regulatory mechanisms in bone. The role of miRNA in bone metabolism, development, strength and disease has been described in recent, prominent reviews [5,6,7]. We present here several examples of regulatory relationships encompassing miRNA and its target mRNA. Most of them are involved in collagen maturation in skeletal and non-skeletal tissues. We suggest that the same regulatory relationship as in non-skeletal cells may also occur in bone cells.

## 2. Collagen Formation

### 2.1. Collagen Expression

#### 2.1.1. *COL1A1* Expression Regulation by miRNAs in Skeletal Tissues

Several types of miRNA regulate *COL1A1* expression in skeletal tissues. The mechanism of collagen gene expression was described in detail in a review by Karsenty and Park [8]. Type I collagen is a heterotrimer molecule consisting of two α 1 chains and one α 2 chain encoded by genes placed on the 17 and 7 chromosomes. Two types of mRNA and polypeptide chains are synthesised in a 2:1 ratio. Sp1 (Specificity protein 1), Dlx5 (distal-less homeobox 5) and Runx2 (Runt-related transcription factor 2) transcription factors regulate the expression of Col1a1 [9,10,11]. 

Activation of the toll-like receptor 4/nuclear factor-kappa B (TLR4/NF-κB) pathway in the intervertebral disc induces pro-inflammatory cytokines, which stimulate the expression of miR-625-5p [12]. An increase in miR-625-5p targeting *COL1A1* mRNA results in a decrease in the *COL1A1* transcript and eventually contributes to the pathological process of intervertebral disc degeneration [12].

#### 2.1.2. Regulation of *COL1A1* Expression by miRNAs in Non-Skeletal Tissues

The miR-196a level is lowered in keloid tissues, and type I collagen is highly deposited in keloid tissues [13]. miR-196a could thus be a new therapeutic target for keloid lesions [13]. Keloids spread beyond the borders of the initial injury and do not regress, while hypertrophic scars are contained within the site of injury and may regress over time [14]. Hypertrophic scarring is a severe disease that results from unusual wound healing. Collagen type I could promote hypertrophic scar formation, and the expression of *COL1A1* in scar tissue was markedly higher than that in normal tissue. By targeting *COL1A1*, increased miR-98 represses the proliferation of hypertrophic scarring [15]. Another study demonstrated upregulated sponging RNA H19 expression in keloid tissue fibroblasts and downregulated miR-29a expression [16]. miR-29a represses type I collagen by interacting with the 3’-untranslated region (UTR) of *COL1A1* mRNA [16]. Consequently, miR-29a decreases fibroblast proliferation and allows for the development of keloid-targeted treatments [16]. Collagen production stimulated by transforming growth factor β (TGF-β) is activated in hepatic stellate cells involved in liver fibrosis. miR-29a exhibits an anti-fibrotic effect that directly suppresses the expression of *COL1A1* and induces apoptosis [17].

*COL1A1* expression diminished by miR-513b-5p affects the viability of vascular smooth muscle cells. This process might involve the formation and rupture of the intracranial Sirtuin 6 (silencing information regulatory protein 6, SIRT6), a member of the sirtuin family proteins [18], which regulates collagen metabolism and is an established anti-ageing protein. miR-128 inhibits osteoblast differentiation in osteoporosis by downregulating *SIRT6* expression, thus accelerating the development of osteoporosis [19]. miR-378b represses the mRNA expression levels of *COL1A1* via interference with SIRT6 in human dermal fibroblasts [20,21].

Several examples of miRNA described here as *COL1A1* targeting, e.g., miR-29b or miR-133, are also involved in bone homeostasis but target other genes [6]. Since the regulation of collagen type I by miR-29b or miR-133 was described in other tissues, and these miRNAs are co-expressed in bone, collagen may possibly be regulated in bone by the same miRNAs.

#### 2.1.3. *COL1A1* and miRNAs in the Development of Neoplasms

Bone is the most common target organ for high-grade metastatic prostate cancer [22]. Tumour growth in bone is the result of crosstalk between tumour and bone cells. Cancer-derived exosomes participate in osteoclast differentiation and increase osteolysis [23]. miRNAs are delivered by these exosomes and play a key role in bone homeostasis [23]. Among these delivered miRNAs, miR-92a-1-5p downregulates *COL1A1* expression, which promotes osteoclast differentiation and inhibits osteoblastogenesis [23].

Increased *COL1A1* expression has been observed in cancer cells, and collagen type I contributes to the differentiation and metastatic abilities of human cancer cells [56,57]. Several miRNAs that interact with *COL1A1* mRNA could be classified as suppressor miRNAs. miR-129-5p suppresses gastric cancer cell proliferation, migration and invasion by selectively inhibiting *COL1A1* [24]. The study of miRNA provides new therapeutic targets for the clinical treatment of gastric cancer [24]. *COL1A1* is also a target gene of miR-133b. An increase in miR-133b inhibits the migration of gastric cancer cells [25]. The assessment of miR-133b expression may have a significant impact on the prognosis of gastric cancer patients. An overexpression of closely related miR-133a-3p (difference with miR-133b only in 3′-terminal A → G) suppresses the proliferation, invasion and mitosis of oral and oesophageal squamous cell carcinoma [26,27]. miR-133, shown to also target *RUNX2* mRNA, with a consequent reduction in *RUNX2* mRNA or its translation, determines the inhibition of osteogenesis in vitro and leads to bone loss in vivo by inducing osteoclastogenesis [7]. miR-133 upregulation results in oestrogen-deficiency-induced osteoporosis and increases the numbers of monocytes in osteoporotic post-menopausal women [58].

In breast cancer, miR-196b-5p inhibits *COL1A1* mRNA by transcript downregulation [28,29]. The closely related miR-196a (difference with miR-196b-5p only in a single nucleotide C → A) directly binds to the 3’-UTR of *COL1A1* [13]. 

### 2.2. Intracellular Stages of Collagen Formation

The collagen type I is post-translationally modified in several steps: hydroxylation of proline and lysine residues, glycosylation of some hydroxylysine residues and the formation of intra-chain and inter-chain disulphide bonds in the terminal peptide (Figure 1).

#### 2.2.1. Hydroxylation of Proline and Lysine Residues, Glycosylation of Some Hydroxylysine Residues

Collagen hydroxylation begins in the endoplasmatic reticulum (ER), where prolyl and lysyl hydroxylases convert proline into hydroxyproline or lysine into hydroxylysine, respectively. Three specific enzymes mediate hydroxylation: prolyl 4-hydroxylase, prolyl 3-hydroxylase and the family of lysyl hydroxylases [59].

The hydroxylation of proline residues on collagen, catalysed by collagen prolyl 4-hydroxylase (C-P4H), is essential for the stability of the collagen triple helix. C-P4H is an α2β2 tetramer consisting of three isoenzymes differing in the catalytic α-subunits, which are encoded by *P4HA1*, *P4HA2* and *P4HA3* genes and a β-subunit, which is encoded by a single gene, *P4HB* [60]. The expressions of *P4HAs* are regulated by multiple cellular factors, including miRNAs [60]. miR-124-3p inhibits the collagen synthesis of vascular smooth muscle cells (VSMC) by directly targeting *P4HA1* [32]. miR-122 plays an important role in negatively regulating collagen production in haematopoietic stem cells [61]. miR-122 is significantly upregulated in osteoclasts from osteoporotic patients and showed a significant sensitivity and specificity in distinguishing osteoporotic patients [62].

*P4HA2* enhances collagen deposition in the liver in vivo and in vitro, leading to liver fibrosis and liver cancer progression. miR-30e, targeting *P4HA2* mRNA, controls collagen production [63].

Proline hydroxylation by procollagen-lysine, 2-oxoglutarate 5-dioxygenase (PLOD) is indispensable for cross-linking and maintaining the mature collagen network and for the formation of the tight triple-helical structure of collagen [64,65]. The reduced form of vitamin C—ascorbic acid—and α-ketoglutarate function as cofactors for these reactions [66]. Three lysyl hydroxylases (LH1, LH2 and LH3) are identified, encoded by *PLOD1*, *PLOD2* and *PLOD3* genes. The expression of *PLOD*s is regulated by multiple cytokines, transcription factors and miRNAs [67]. miR-34c inhibits *PLOD1* expression in osteosarcoma at both mRNA and protein levels [35]. miR-34c, by targeting and suppressing special AT-rich sequence binding protein 2 (SATB2), has negative effects on bone development [68]. miR-34a was found to suppress the osteoblast differentiation of human MSCs in vitro and reduce the formation of bone following the subcutaneous transfer of hMSC-loaded ceramic beads in SCID mice [69].

miR-124 and miR-26 targets *PLOD2* mRNA in regulating the malignant behaviours of laryngeal and renal carcinoma cells and is increased in osteoporotic patients’ blood [36,37,62]. In osteogenesis in vitro miR-26 regulates glycogen synthase kinase 3 β gene *GSK3B* [70]. *PLOD3* is held by miR-663a, modulating collagen IV secretions in physiological conditions and in response to ER stress [38].

#### 2.2.2. Procollagen Glycosylation

Hydroxylated lysyl residues allow galactosyltransferase (GLT25D1 and GLT25D2, also known as COLGALT1 and COLGALT2, and glucosyltransferase to transfer glucose and galactose to the hydroxylysine residues of procollagen [39,40]. The gene encoding the 1-2 glucosyltransferase enzyme remains unknown [40]. miRNA regulates glycosyltransferases, but *GLT25D1* and *GLT25D2* regulatory miRNAs have not been identified [41]. Glycosylation is engaged in the folding and stability of procollagen and the prevention of inter-chain cross-linking after hydroxylation and glycosylation. C propeptides from two α chains and one β chain are associated with forming intrachain and interchain disulphide bonds [71].

Many patients diagnosed with low bone mineral mass do not experience risky fractures [72]. Enzymatic (glycosylation) and non-enzymatic (glycation) are the two types of post-translational modifications of extracellular matrix proteins and collagen proteins that influence bone quality at the macro-, micro- and nanoscale. Protein in connection with carbohydrates participates in mineralisation, microdamage and the mechanical properties of bone [73].

#### 2.2.3. Formation of Intra-Chain and Inter-Chain Disulphide Bonds in Terminal Peptides

Protein disulphide isomerase (PDI) encoded by *P4HB* catalyses the formation of intra-chain and inter-chain disulphide bonds in the C-propeptide regions of each collagen peptide [33,74,75]. PDI is the β-subunit of α_2_β_2_ heterotetramer of collagen prolyl 4-hydroxylase, and acts as a chaperone, stabilising the functional conformation of collagen [76]. The miR-210 P4HB target is highly downregulated in chemotherapeutic agent temozolomide-resistant glioblastoma multiforme cells [34]. These findings suggest new directions for *COL1A1* studies to examine if miR-210 regulates collagen formation in bone.

#### 2.2.4. Procollagen Triple Helix Formation

Hsp-47 (heat shock protein 47, encoded by *SERPINH1*) is a collagen-specific chaperone essential for correct procollagen triple helix folding in the ER [77,78]. Hsp-47 also prevents the local unfolding of procollagen and aggregation [77,78]. Mutations in the Hsp-47 gene lead to osteogenesis imperfecta [77,78]. In Hsp-47-negative cells, the folding of procollagen is impaired, and due to the malformation of its triple helix, it is more sensitive to proteolysis [77,78]. Hsp47 mRNA contains a binding site for miRNA-29b in the 3′-UTR [43]. The simultaneous downregulation of miR-29a and upregulation of Hsp47 has been reported in cervical squamous cell carcinoma [42]. In breast cancer and glioma cells, the expression of miR-29b suppresses malignant phenotypes by reducing Hsp47 and collagen deposition [43,79]. The downregulation of miR-29b and upregulation of Hsp47 were observed during wound healing [80]. These findings bear translational possibilities, e.g., that targeting miR-29b/Hsp47 might be a strategy to reduce scar formation [80]. Since the miR-29 family plays an important role in regulating osteoclasts, osteoblasts and bone marrow mesenchymal stem cells (BM-MSC), it is likely that Hsp47 is also a target of miR-29 in bone [6]. The triple helix formation is secreted extracellularly via the Golgi apparatus, and probably also the large coat protein complex II (COPII) and TANGO1 [81,82].

### 2.3. Extracellular Stages of Collagen Formation

#### 2.3.1. Cleavage of Amino- and Carboxy Propeptides

Extracellularly, procollagen undergoes several additional modifications. Specific N- and C- propeptides remove amino- and carboxy-terminal propeptides, respectively [83]. Procollagens are modified by the amino-terminal propeptides disintegrin and metalloproteinase with thrombospondin motifs (ADAMTS2, -3 and -14), and the carboxy-terminal propeptides bone morphogenetic protein 1 (BMP1)/Tolloid-like families, respectively [84]. BMP1 also cleaves and activates the lysyl oxidase (LOX) precursor, the enzyme catalysing the formation of covalent collagen cross-links, an essential process for fibril stabilisation.

The expression of multiple genes involved in the synthesis and deposition of extracellular matrix in human trabecular meshwork cells is negatively regulated by miR-29b [45]. *COL1A1*, *BMP1* and *ADAM12* mRNAs were identified as direct targets of miR-29b. miR-194 expression was found to be strongly negatively associated with metastasis in clinical specimens of non-small cell lung cancer [46]. miR-194 directly targets BMP1 [46]. The resulting downregulation of BMP1 leads to the suppression of TGF-β activity and thus the downregulation of the expression of critical oncogenic genes (matrix metalloproteinases MMP2 and MMP9) [46]. The progression of hepatocellular carcinoma was negatively regulated by miR-29c and promoted by BMP1 [47]. miR-29b overexpression has been implicated in osteoblast differentiation of cell line MC3T3-E1 [85]. Thus, miR-29b is a crucial regulator of the osteoblast phenotype by targeting anti-osteogenic factors and modulating bone extracellular matrix proteins [85]. It remains to be seen if BMP1 is also a target of miR-29b in bones.

#### 2.3.2. Aggregation of Collagen Fibres

Collagen fibres aggregate spontaneously due to a self-assembly mechanism, and the shift between fibres is kept to 1/4 of the length [48]. The mechanical strength of bone is highly dependent on the formation of covalent cross-links within collagen fibrils. The cross-link is initiated by the enzymatic action of LOX [43,84]. LOX catalyses the oxidative deamination of epsilon-amino groups of lysyl and hydroxylysine residues located in telopeptide domains to aldehydes (allysine and hydroxylysine) [2,86,87]. Reactive aldehydes initiate non-enzymatic condensation reactions to form covalent bonds with lysine or hydroxylysine residues, which initiates collagen cross-linking [2]. LOX requires copper and several cofactors: pyridoxal phosphate (vitamin B6) and tyrosyl-lysine quinone [88,89]. Active vitamin D3 (calcitriol) increases the expression of LOX [90]. Thus, a vitamin B6 deficiency, or a copper deficiency lowering LOX activity, impairs enzymatic cross-linking and causes a decrease in bone strength [2,86]. Another possibility resulting from the deficient action of lysyl oxidase causing defects in enzymatic collagen cross-linking is Menkes disease [91]. *LOX* expression is regulated by miR-27, miR29a/b, miR30a/b and miR-142.

Lysyl oxidase is upregulated and functions as an essential factor during bone morphogenetic protein 4-induced murine mesenchymal stem cells (C3H10T1/2) differentiation to adipocytic cell lineage. miR-27, by targeting *Lox* mRNA, represses adipogenic lineage differentiation [49]. Additionally, in mesenchymal stem cells, miR-27 determines the inhibition of osteoblastic differentiation by affecting a negative regulator of the Wnt signal [7,92]. miR-27, by targeting *SP7* gene (osterix, *OSX*), suppresses the osteogenic differentiation of maxillary sinus membrane stem cells [93]. miR-27a is one of the most strongly downregulated miRNAs in larger, post-menopausal, osteoporotic women and healthy, pre-menopausal women [92].

miR-29a has been shown to exert a hepatoprotective effect on hepatocellular damage and liver fibrosis [50]. In hepatocellular carcinoma, miR-29a-3p exerts inhibitory activity directly, binding to the 3’-UTR of *LOX* [50]. A low expression of miR-29a and high expression of *LOX* indicates poor survival [50]. miR-29a promotes osteogenesis and suppresses histone deacetylase 4 (*HDAC4*), indicating that decreasing miR-29a may be feasible in the management of osteoporosis [94]. miR-29b is also implicated in the pathogenesis and progression of liver fibrosis by regulating the post-translational processing of extracellular matrix (ECM) and fibril formation [43]. In rat hepatic stellate cells, miR-29b targeting the 3’-UTR sequence in LOX mRNA leads to abnormal collagen structure [43].

Direct regulation of *LOX* by miR-30a was confirmed in human keratinocytes. miR-30a overexpression strongly impaired epidermal differentiation; a significant increase was also observed in the level of apoptotic cells overexpressing miR-30a in the epidermis [51]. LOX was involved in primary tumour formation, and the establishment of metastases was verified as the downstream target of miR-30c-2-3p gastric cancer [52].

miR-142-3p may be developed and tested with other LOX inhibitors to overcome chemoresistance in triple-negative breast cancer [53]. Chemoresistance is a trait of triple-negative breast cancer, but inhibiting LOX by miR-142-3p reduces collagen cross-linking and fibronectin assembly, increases drug penetration and causes the induction of apoptosis and re-sensitisation to chemotherapy [53].

#### 2.3.3. Crosslinking

Finally, cross-links are formed between and within the chains, which is a significant post-translational modification of collagen, providing the neighbouring collagen molecules with a stable structure and toughness in the bone [2]. Cross-links can be created both enzymatically and non-enzymatically [2,95]. LOX regulate enzymatic cross-link formation [2]. *LOX* gene expression is regulated by several miRNA families: miR-27, miR29, miR-30 and miR-142 (see Table 1 and Figure 2) [43,49,50,51,52,53,54,55]. The non-enzymatic process is pathological, and it is mainly based on non-enzymatic glycation [2,95,96,97]. During the ageing process or some diseases (e.g., diabetes), the non-enzymatic cross-links, also known as advanced glycation end products, accumulate in collagen and cause reduced plasticity, loss of toughness, lower bone strength and increased fracture risk [2,91,97,98,99].

## 3. Collagen Interactions with Non-Collagenous Proteins and Their Impact on Bone Quality

Collagen fibres function as a scaffold for bone cells, and they are the main components providing bone tissue with strength and elasticity. Any disorders during the process of collagen formation affect the mechanical properties of the bone. The bone owes its remarkable mechanical features and resistance to fracture to its hierarchical composite structure. The involvement of miRNA in bone remodelling induced by mechanical forces was recently described [5]. Despite collagen and hydroxyapatite being the most abundant, various non-collagenous matrix components, such as proteoglycans and glycoproteins, play a crucial role in the mechanical properties of bone tissue (Table 2). The matrix of bone comprises collagen type I stabilised by water and non-collagenous proteins (NCPs). NCPs can accumulate Ca^2+^ ions in their proximity and bind collagen [100]. Interactions between collagens and NCPs significantly impact bone strength and resistance to fracture [101]. NCPs attach cells to the bone matrix, and they are essential for remodelling, which alters the material and whole bone quality [101]. This work focuses on a few significant NCPs and their influence. One of the most important NCPs is osteopontin (OPN), which is produced by osteoblastic and osteoclastic cells at high levels [102].

### 3.1. Osteopontin

OPN is a phosphorylated glycoprotein, and it plays a role in the mineralisation, organisation and deposition of the extracellular matrix in bone [103]. OPN provides the cohesion of collagen fibrils and prevents the collapse of its structures by mineralisation [104]. After an intrafibrillar mineral is formed, OPN may alter the growth habit of extrafibrillar hydroxyapatite crystals through the binding of phosphate and carboxylate groups to minerals [101]. First, the intrafibrillar collagen spaces are mineralised; second, an extrafibrillar mineral coating is formed [104]. *Opn* knock-out mice had no macroscopic alterations in their bones; however, they showed a high mineral content with larger, more perfect crystals, which reduces fracture properties, such as strength and flexibility [104]. Tests showed a greater decrease in hardness or elastic modulus in younger animals than in adult mice [101]. Hence, OPN promotes the early stages of mineralisation, osteoclast activity and the adherence of osseous cells [102,104,105]. *Opn*-deprived mice can still mineralise bone; therefore, there must also be other NCPs that play a role in mineralisation [101].

Interestingly, miR-127-5p inhibited the proliferation of chondrocytes through a decrease in *OPN* expression [106]. miRNA-127-5p mimics suppressed *OPN* production and the activity of a reporter construct containing the 3’-UTR of human *OPN* mRNA [106]. The levels of miR-4262 were significantly decreased, and the level of OPN was increased considerably in osteosarcoma specimens compared to the paired adjacent non-tumour tissue [107]. miR-4262 overexpression inhibited OPN-mediated cell invasion, highlighting miR-4262 as an intriguing therapeutic target to prevent osteosarcoma metastases [107]. The role of miR-127-5p and miR-4262, which target the 3’-UTR of *OPN* in bone, has not yet been reported.

### 3.2. Thrombospondin-2

Thrombospondin-2 (TSP2, encoded by *THBS2*) belongs to a family of five extracellular proteins, promoting osteoblast lineage progression [108]. Thsp2-knockout mice had reduced matrix collagen levels compared to the wild types [109]. TSP2 incorporates collagen into the insoluble cross-linked bone matrix, although it does not affect total collagen production [110]. TSP2 is an osteogenesis regulator, expressed in bone progenitor cells, such as MC3T3, acting in an autocrine manner and promoting osteoblast differentiation and bone deposition [111]. However, TSP2 is also produced by non-skeletal cells, such as trophoblasts and colon cancer cells, where the *THBS2* gene is expressed and regulated by miRNAs.

miR-221-3p negatively regulated the expression of *THBS2* in human first-trimester placenta (HTR-8/SVneo) cells [111]. In vitro functional assays have revealed that miR-221-3p promotes trophoblast growth, invasion and migration partly via targeting *THBS2* [111]. In terminal osteoblasts, miR-221 attenuates differentiation through Dkk2 mRNA targeting. *THBS2* has been predicted and experimentally verified as a direct target of miR-93-5p [112]. The promotion function of miR-93-5p on cervical cancer operates by targeting the *THBS2*/matrix metalloproteinases signalling pathway. miR-93-5p might be a potential therapeutic target for the treatment of cervical cancer [112].

miR-203a-3p binds to the 3’-untranslated region of *THBS2* mRNA, downregulating *THBS2* expression and inhibiting colorectal cancer progression and metastasis [113]. The expression of miR-203a-3p, which serves a tumour-suppressive role, was significantly downregulated in colorectal cancer tissues. miR-203a-3p was determined to target *THBS2* to inhibit colorectal cancer progression and metastasis; thus, miR-203a-3p may be considered a potentially novel approach to treating colorectal cancer [113].

miR-93, miR-203 and miR-221 are also involved in bone homeostasis by targeting *BMP2*, *Smad9* and *RUNX2,* respectively [114,115,116]. Since miR-93, miR-203 and miR-221 targeting *THBS2* was described in other tissues, *THBS2* may possibly be regulated in bone by the same miRNAs.

### 3.3. Biglycan and Decorin

Signalisation of TGF-β has been identified as responsible for the elasticity and hardness of the bone matrix [117]. The miRNA regulation of the TGF-β pathway has already been reviewed [118]. The secretion of TGF-β by bone cells increases minerals in the bone matrix from about 33–42% [117]. While TGF-β is an essential regulator of osteogenesis, proteoglycans are significant regulators of TGF-β [117]. Small leucine-rich proteoglycans, such as biglycan and decorin, are responsible for binding this growth factor. Biglycan and decorin are necessary to maintain an adequate number of mature osteoblasts by modulating the proliferation and survival of bone marrow stromal cells. Biglycan deficiency in bones impacts the number of collagen fibril anomalies, including changes in both fibril size and fibril shape, as well as peak bone mass, and can result in the development of osteopenia [119]. 

Biglycan is the target gene of miR-330-5p in bone marrow stem cells [120]. Importantly, biglycan was able to reverse the regulatory effects of miR-330-5p on the BMP/Smad pathway, alkaline phosphatase activity and mineralisation ability in BM-MSCs [120]. miR-330-5p facilitates osteogenesis in bone marrow stromal cells through the biglycan-induced BMP/Smad pathway, thus alleviating the progression of osteoporosis. Osteoporosis mice with in vivo knockdown of miR-330-5p presented higher bone mineral density and BV/TV than controls [120].

Osteoblastic differentiation is regulated by TGF-β signalling molecules, such as TGF-β type I receptor (TβR-I/Alk5). miR-181a promotes osteoblastic differentiation by repressing TGF-β type I receptors [121]. Furthermore, blocking miR-181b reversed TGF-β1-induced decorin downregulation [122]. On the other hand, by blocking miR-181b, it was proven that decorin is an miR-181b target gene in dermal fibroblasts [122]. The deletion of miR-181 in mice revealed a potential role for this miRNA in controlling growth plate and bone development, as well as the enhancement of osteogenesis in vitro [123].

### 3.4. Osteonectin

Osteonectin, also called ‘secreted protein acidic and rich in cysteine’ (encoded by *SPARC*, BM-40), is produced by osteoblasts. Osteonectin is responsible for bone density and trabecular volume, with effects on microarchitecture, the most abundant non-collagenous matrix protein in bone. The loss of osteonectin in the bone results in decreased trabecular bone [124]. Osteonectin modifies the balance between bone formation and bone resorption, affecting both sides of the remodelling process. Furthermore, it initiates active mineralisation, and complexes of osteonectin–collagen bind apatite crystals and free calcium ions [125]. The proximal region of the mouse osteonectin 3′-UTR contains a well-conserved, dominant regulatory motif that interacts with miRs-29a and miR-29c. Wnt signalling, which is increased during osteoblastic differentiation, induces the expression of miR-29 [126].

## 4. Interactions between Collagen and the Mineral Phase in Bone

Here, we will focus on the organisation of mineralised collagen fibrils and the interactions of mineral compounds with collagen. Phenomena that occur on the nanoscale level are fundamental to understanding the structure of bone. Mineral compounds in bone belong to the calcium phosphate family. Precisely, it is a type of apatite [Ca_5_(PO_4_)_3_(F, Cl, OH)], the form of which resembles hydroxyapatite [Ca_10_(PO_4_)_6_(OH)_2_] but varies in its contents [100]. The presence of substantial carbonate CO_3_^2−^ together with some cations (H^+^, Na^+^, Mg^2+^, K^+^) and anions (F^−^, Cl^−^) proves the specificity of the apatite [100]. The mineral occurs in bone in the form of flake-like nanosized platelets, whose length ranges from 15–200 nm, thickness from 1–7 nm and width from 10–80 nm [100,132,133]. According to classical models, the apatite crystals lie between collagen fibrils [134,135]. It has been postulated that interactions between collagen and the mineral phase provide bone with appropriate mechanical properties.

When implemented into the fibril, the mineral phase causes a decrease in length and induces internal stress. On the other hand, the presence of the mineral phase improves the stiffness, strength, extensibility and toughness of the fibril.

Alkaline phosphatase (gene *ALPL*) activity alters the Pi/PPi ratio in the bone microenvironment to favour bone mineralisation [136] and is an agent in mineralisation regulated by miRNA. Skeletal ALPL is anchored to membrane inositol-phosphate on the outer surface of osteoblasts [137]. ALPL-mediated hydrolysis of PPi has two implications. First, it reduces the amount of PPi in the bone microenvironment. Second, it increases the amount of mineralisation promoting Pi ions by liberating them from PPi [136]. Lower protein expression levels were observed using Western blotting, confirming that ALPL is directly targeted by hsa-miR-149 [128]. Inhibition of miR-99a-5p in MC3T3 pre-osteoblastic cells promoted osteogenic differentiation, whereas its overexpression suppressed the levels of osteogenic-specific ALPL expression [129]. miR-9-5p could bind directly to the 3’-UTR of ALPL and inhibit the expression of ALPL. Circular RNA circSIPA1L1 upregulates ALPL by targeting miR-204-5p and promoting the osteogenic differentiation of stem cells from the apical papilla [131].

There are many lines of evidence indicating that mechanical stress regulates bone metabolism and promotes bone growth. BMP, Wnt, ERK1/2 and OPG/RANKL are the main molecules that regulate the effects of mechanical loading on bone formation. Recently, miRNAs were found to be involved in mechanical stress-mediated bone metabolism [138].

However, Wolff’s law of bone remodelling states that bone adapts to its loads [139]. Some concepts hold that miRNAs may be critical for exercise-induced bone remodelling [140].

## 5. Conclusions

The quantity and spatial organisation of collagen, non-collagenous proteins, polysaccharides and minerals determine the mechanical properties of bone. This review discusses the epigenetic regulation of collagen and non-collagenous protein by miRNA in numerous tissues and diseases. Several collagen-regulating miRNAs are co-expressed with collagen in bone. The functionality of miRNA–collagen relationships has not yet been proven in osteoblasts, osteocytes and osteoclasts. We conclude that the regulatory mechanisms of these collagen and target miRNAs could also regulate bone metabolism, which is a matter for future research.

## Figures and Tables

**Figure 1 ijms-23-00091-f001:**
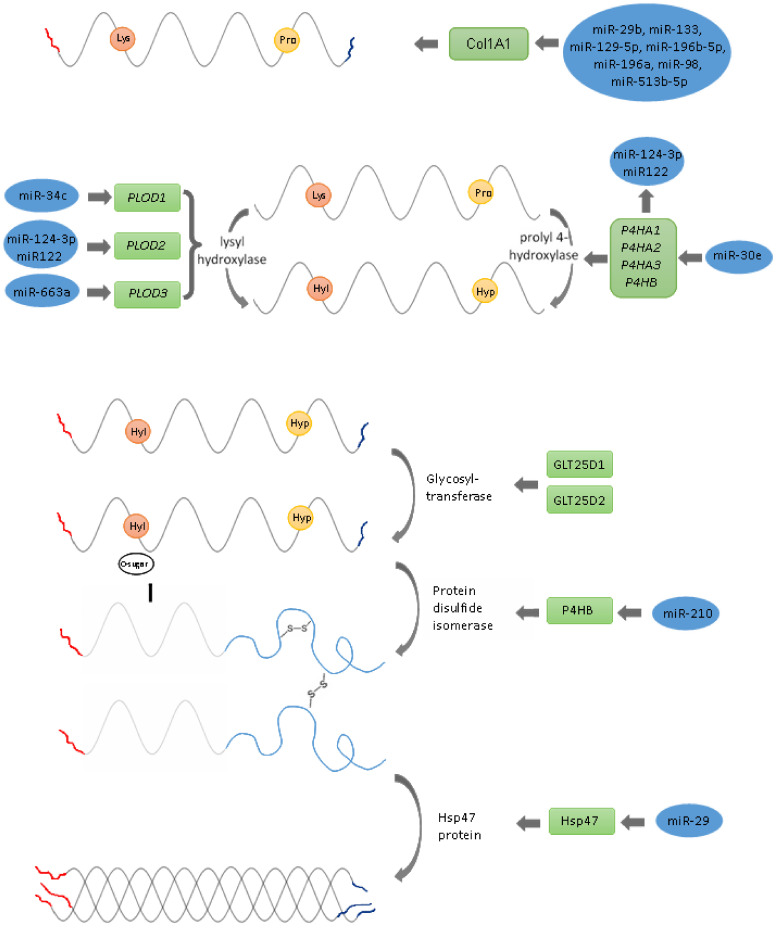
Schematic presentation of the key steps of collagen synthesis in the cell.

**Figure 2 ijms-23-00091-f002:**
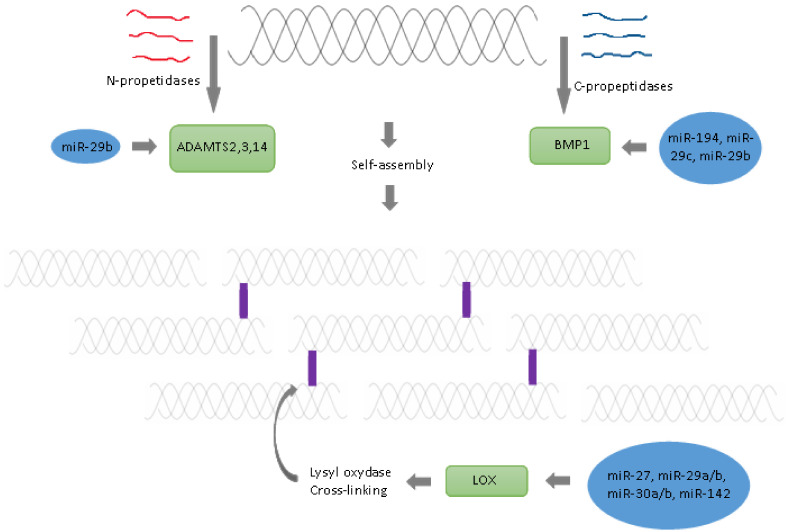
Schematic representation of the key steps of collagen synthesis in the extracellular matrix.

**Table 1 ijms-23-00091-t001:** The enzymes involved in the crucial stages of collagen formation.

	Intracellular Stage of Collagen Formation	Enzyme	Gene	miRNA	References
1.	*COL1A1* gene expression	−	*COL1A1*	miR-625-5p, miR-92a-1-5p, miR-129-5p, miR-133b, miR-133a-3p, miR-196b-5p, miR-196a, miR-98, miR-29a, miR-513b-5p	[12,13,15,16,17,18,23,24,25,26,27,28,29]
2.	Hydroxylation of proline residues	Prolyl,4- hydroxylase	*P4HA1*, *P4HA2*, *P4HA3*, *P4HB*	miR-124-3p, miR-122, miR-30e, − miR-210	[30,31,32] [33,34]
3.	Hydroxylation of lysine residues	Lysyl hydroxylase	*PLOD1*, *PLOD2*, *PLOD3*	miR-34c, miR-124, miR-26, miR-663a	[35,36,37,38]
4.	Glycosylation of some hydroxylysine residues	Glycosyltransferase	*GLT25D1* *GLT25D2*	−	[39,40,41]
5.	Formation of intra-chain and inter-chain disulphide bonds in terminal peptides	Protein disulphide isomerase	*P4HB*	miR-210	[33,34]
6.	Procollagen triple helix formation	Hsp-47 protein	*SERPINH1*	miR-29b, miR-29a	[42,43,44]
	**Extracellular stage of collagen formation**	**Enzyme**	**Gene**	**miRNA**	**Sources**
7.	Cleavage of amino propeptides	Specific N-propeptidases	*ADAM2*,*3 and 14*	miR-29b	[45]
8.	Cleavage of carboxy propeptides	Specific C- propeptidase	*BMP1*	miR-194, miR-29c, miR-29b	[45,46,47]
9.	Aggregation of collagen fibres	Self-assembly	*−*	−	[48]
10.	Crosslinking Oxidative deamination of epsilon-amino groups of lysyl and hydroxylysine residues located in telopeptide domains to aldehydes	Lysyl oxidase	*LOX*	miR-27, miR29b, miR29a, miR-30a, miR-30b, miR-30b-2-3p, miR-142-3p	[43,49,50,51,52,53,54,55]

**Table 2 ijms-23-00091-t002:** Regulation of non-collagenous proteins by miRNA.

Protein	Function	Gene	miRNA	References
TGF-β	Regulates the elastic modulus and the hardness of the bone	*TGFB1*	See review	[118]
Decorin	Models the effects of TGF-β, participates in fibrogenesis of collagen, prevents the mineralisation of collagen [30]	*DCN*	miR-181b	[122]
Biglycan	Inhibits the effects of TGF-β	*BGN*	miR-330-5p	[120]
Osteonectin	Responsible for bone density	*SPARC*	miRs-29a and -29c	[127]
Osteopontin	Crucial for the deformability and resistance of bones to fractures, provides strong adhesion for hydroxyapatite and bone sialoprotein I (BSP-1 or BNSP)	*OPN*	miRNA-127-5p miR-4262	[106,107]
Thrombospondin-2	Incorporates collagen into the insoluble cross-linked bone matrix	*THBS2*	miR-221-3p miR-93-5p miR-203a-3p	[111,112,113]
Alkaline phosphatase	Alters the Pi/PPi ratio in the bone microenvironment to favour bone mineralisation	*ALPL*	hsa-miR-149, miR-99a-5p MiR-9-5p miR-204-5p	[128,129,130,131]

## Data Availability

Not applicable.

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
