# Peer review of "The Regulation of Collagen Processing by miRNAs in Disease and Possible Implications for Bone Turnover"

_ijms, 2021, doi:10.3390/ijms23010091_

Round 1
Reviewer 1 Report
Lehmann et al. here provide an overview of the miRNA-based collagen regulation of bone formation and metabolism. This topic represents an excellent example of how miRNA intervention may regulate tissue development and differentiation. The authors here start describing the role of miRNA in collagen formation, giving details on the intracellular and extracellular regulation of this process and on the interactions with proteins involved in bone metabolism.
Overall the manuscript is well written and provides mechanistic insights of main biological processes underpinning this topic.
I have some suggestion to improve the flow and the readability of this review.
- First of all I think the authors should put some efforts in the visualization of the main concepts. I suggest introducing a couple of figures showing the steps of collage formation and homeostasis highlighting the key miRNA and the key mRNA targets.
- Although not within the purpose of this review, I also think that Readers may find useful to see an introductory paragraph focused on what is a miRNA and how it regulates post-transcriptional processes.
- Are some of these miRNA operative in other compartments? If possible, it would be valuable to characterize (eventually in an additional table) the specificity of each single miRNA listed in tissue development and differentiation.
- Another important concept is the role of these processes in osteoporosis and other bone disorders. I suggest adding a paragraph focused on the pathophysiology of miRNA regulation in pathologic bone metabolism: this would be essential to give a full perspective on the clinical meaning of this mechanisms.
Author Response
"Please see the attachment."

Reviewer 2 Report
In this review, Lehmann et al. focused on miRNAs which mainly regulate collagen formation and processing. The authors summarized implication of these miRNAs in collagen maturation, and also discussed their roles in bone turn over. I’d like to raise several points which may require revisions.
- It is better to make introduction. The first three paragraphs in section 1. Collagen formation sounds introduction of this review. I suggest to arrange these paragraphs and make introduction.
- Page 2 line 56; this sentence is illegible. Is this single sentence? Or is “COL1A1 expression regulation by miRNA” a heading?
- In several points of citation, detailed explanation is required. For example, in page 2 line 57-58, “The mechanism of collagen gene expression has been deeply described in a review by….[7]”. I suggest to add short description about collagen gene expression mechanisms.
- Page 4 line 112-115: this sentence seems introduction of the next section 2. Intracellular stages of collagen formation.
- Acronym should be spell-out only once where it is shown the first time. For example, page 5 line 193 lysyl oxidase (LOX) should be described LOX in the following. It does not have to spell-out in page 6 line 215 or skip acronym in line 226.
- Page 7 line 258-260; these sentences seem introduction of the next section 4. Collagens interactions with non-collagenous proteins and their impact on bone quality.
- The authors claim “collagen-miRNAs regulatory loops”, but I do not quite understand what is the “collagen-miRNAs regulatory loops”.
- In whole, this review is indecisive in terms of structure and essential point of topics, which is due in part to grammatical confusion.
Author Response
"Please see the attachment."

Round 2
Reviewer 2 Report
The authors replied to my questions, and the contents seems improved. However, I have few things to point out.
- I have read carefully, and also read the reference by Zhang et al. However, still quite not clear about “collagen-miRNA regulatory loops”, that the authors claimed. The authors have to add explanation of the “loops” in the main text. If it was already added, please describe where it is (page No. and line No.). In addition, please add clear scheme of the “loop” in Figures with detailed figure legends.
- page 1, line 42: what is “miR-mRNA regulatory loops”?
- page 4, line 113-115: I do not understand this sentence.
- The manuscript has to go through very extensive English editing (language and style).
Round 3
Reviewer 2 Report
The manuscript was greatly improved. I could understand the contents well. I have two suggestions before accepting this manuscript.
- Abstract: this paragraph starts with "This article describes several recent ...", then, there is a sentence in the middle of the paragraph "In the current review, we show..." These sentences confuse readers to understand which is the main message of this review article. I suggest to simply delete the words "In the current review" (in line 24).
- 2.3.3. Crosslinking: this paragraph did not mention about miRNAs. If no publication of involvement of miRNA in collagen crosslinking process has been reported, it should be written here, like "Involvement of miRNA in this process is not yet determined".
